# Review of Indications of FDA-Approved Immune Checkpoint Inhibitors per NCCN Guidelines with the Level of Evidence

**DOI:** 10.3390/cancers12030738

**Published:** 2020-03-20

**Authors:** Raju K. Vaddepally, Prakash Kharel, Ramesh Pandey, Rohan Garje, Abhinav B. Chandra

**Affiliations:** 1Depratment of Hematology/Medical Oncology, Yuma Regional Medical Center, Yuma, AZ 85364, USA; abhinavbck@hotmail.com; 2Department of Internal Medicine, Geisinger Medical Center, Danville, PA 17822, USA; pkharel@geisinger.edu; 3Department of Internal Medicine, MD Anderson Cancer Center, Houston, TX 77030, USA; pandey.rmh@gmail.com; 4Department of Hematology/Medical Oncology, University of Iowa, Iowa City, IA 52242, USA; rohan-garje@uiowa.edu

**Keywords:** FDA, NCCN, immunotherapy, CTLA-4, PDL-1, checkpoint inhibitors, cancer, nivolumab, pembrolizumab, ipilimumab, atezolizumab, durvalumab, avelumab, cemiplimab, T-cells activation

## Abstract

Cancer is associated with higher morbidity and mortality and is the second leading cause of death in the US. Further, in some nations, cancer has overtaken heart disease as the leading cause of mortality. Identification of molecular mechanisms by which cancerous cells evade T cell-mediated cytotoxic damage has led to the modern era of immunotherapy in cancer treatment. Agents that release these immune brakes have shown activity to recover dysfunctional T cells and regress various cancer. Both cytotoxic T-lymphocyte-associated protein 4 (CTLA-4) and Programmed Death-1 (PD-1) play their role as physiologic brakes on unrestrained cytotoxic T effector function. CTLA-4 (CD 152) is a B7/CD28 family; it mediates immunosuppression by indirectly diminishing signaling through the co-stimulatory receptor CD28. Ipilimumab is the first and only FDA-approved CTLA-4 inhibitor; PD-1 is an inhibitory transmembrane protein expressed on T cells, B cells, Natural Killer cells (NKs), and Myeloid-Derived Suppressor Cells (MDSCs). Programmed Death-Ligand 1 (PD-L1) is expressed on the surface of multiple tissue types, including many tumor cells and hematopoietic cells. PD-L2 is more restricted to hematopoietic cells. Blockade of the PD-1 /PDL-1 pathway can enhance anti-tumor T cell reactivity and promotes immune control over the cancerous cells. Since the FDA approval of ipilimumab (human IgG1 k anti-CTLA-4 monoclonal antibody) in 2011, six more immune checkpoint inhibitors (ICIs) have been approved for cancer therapy. PD-1 inhibitors nivolumab, pembrolizumab, cemiplimab and PD-L1 inhibitors atezolizumab, avelumab, and durvalumab are in the current list of the approved agents in addition to ipilimumab. In this review paper, we discuss the role of each immune checkpoint inhibitor (ICI), the landmark trials which led to their FDA approval, and the strength of the evidence per National Comprehensive Cancer Network (NCCN), which is broadly utilized by medical oncologists and hematologists in their daily practice.

## 1. Introduction

Cancer is associated with higher morbidity and mortality and is the second leading cause of death in the US. Further, in some nations, cancer has overtaken heart disease as the leading cause of mortality. If this trend continues, by the year 2020, as per the CDC, cancer will become the leading cause of death in the US [1,2]. Identification of molecular mechanisms by which cancerous cells evade T cell-mediated cytotoxic damage has led to the modern era of immunotherapy in cancer treatment. Co-inhibitory T-cell signals, which generally prevent aberrant or chronic activation of the immune mechanism, may have been opted by the cancerous cell to dampen immunity. Agents that release these immune brakes have shown activity to recover dysfunctional T cells and regress various cancer [3,4] [The effectiveness of these agents in cancer treatment and their relatively manageable side effect profiles have established immunotherapy as an alternative modality in cancer treatment. In some solid tumors (e.g., metastatic melanoma and non-small-cell lung cancer), immunotherapy is the first line of treatment.

Immunotherapy was utilized for cancer treatment in the past with limited success. The history of immunotherapy in cancer can be traced back to 1891. William B. Coley injected streptococcal organisms into patients with bone sarcomas [5]. In the 1980s, high-dose interleukin-2 (IL-2) was used successfully in renal cell cancer, producing a prolonged and durable response along with overall survival benefit in a small subset of patients [6,7]. Bacillus Calmette–Guerin (BCG) vaccine, with its ability to induce inflammation, is still being effectively used for the treatment and secondary prevention of non-muscle invasive bladder cancer [8]. Since the FDA approval of ipilimumab (human IgG1 k anti-CTLA-4 monoclonal antibody) in 2011, six more immune checkpoint inhibitors (ICIs) have been approved for cancer therapy. Programmed Death-1 (PD-1) inhibitors nivolumab, pembrolizumab, cemiplimab and Programmed Death Ligand-1 (PDL-1) inhibitors atezolizumab, avelumab, and durvalumab are in the current list of the approved agents in addition to ipilimumab. Multiple agents have been tested successfully either alone or in combination with other agents such as immunotherapy and/or chemotherapy in various malignancies. With more agents under investigation and new immune targets (checkpoints such as BTLA, VISTA, TIM-3, LAG3, and CD47; co-stimulatory molecules such as CD137, OX40, and GITR) being identified, the potential for immunotherapy in cancer treatment is broadening [9,10,11,12,13,14,15,16,17,18,19]. This study aims to summarize the indications of currently approved immunotherapeutic agents with their levels of scientific evidence. In this review paper, we discuss the mechanism of check-point inhibitors (Figure 1), the role of each drug in various solid tumors (Figure 2), the landmark trials which led to their FDA approval, and the strength of the evidence per National Comprehensive Cancer Network (NCCN), which is broadly utilized by medical oncologists and hematologists in their clinical practice.

## 2. Mechanisms of Action

Conventional T cells (Tcon) target tumor cells by two mechanisms (Figure 1). The first mechanism involves an antigen-specific signal through T cell receptors (TCRs) [20]. The second mechanism involves antigen-nonspecific signals mediated by co-signaling receptors—co-stimulatory (act by accentuating T cells responses) or co-inhibitory (act by attenuating T cell response). CD28 is an important co-stimulatory receptor, whereas CTLA-4 and PD-1 are co-inhibitors [21]. Both CTLA-4 and PD-1 play a vital physiological role to put brakes on unrestricted cytotoxic T effector function. CTLA-4 (CD 152) is a B7/CD28 family, and it facilitates immunosuppression by indirectly depleting signaling through the co-stimulatory receptor CD28. As it has a higher affinity for CD80 (B7-1) and CD86 (B7-2), it outcompetes CD28, which leads to a reduction in the release of pro-effector cytokines such as IL-12 and cytotoxic enzymes such as perforin and granzyme B [22,23]. CTLA-4 also mediates the endocytosis of CD80 and CD86 from antigen-presenting cells (APCs), and thus reduces their availability for CD28 [24]. Thereby, this results in an increased activation threshold of T cells, diminishing the immune responses to weak antigens such as self and tumor antigens (TAAs). As a result, CTLA-4 has been a target for therapies to enhance anti-tumor immunity by its blockade with a monoclonal antibody. Ipilimumab is the first and only FDA-approved CTLA-4 inhibitor, given its ability to modualte the immune system to attack melanoma cancer cells [25].

Programmed Death 1 (PD-1) is an inhibitory transmembrane protein expressed on T cells, B cells, Natural Killer cells (NKs), and Myeloid-Derived Suppressor Cells (MDSCs). It binds to the PD-1 ligand (PD-L1; also known as B7-H1) and PD-L2 (B7-H2). Programmed Death-Ligand 1 (PD-L1) is expressed on the surface of multiple tissue types, including many tumor cells and hematopoietic cells; PD-L2 is more restricted to hematopoietic cells. The PD-1–PD-L1/2 interaction directly leads to various diminutive mechanisms such as inhibition of tumor cell apoptosis, peripheral T effector cell exhaustion, and conversion of T effector cells to regulatory T-cells (Treg cells) [26,27]. As such, blockade of the PD-1 pathway can enhance T cell anti-tumor activity and thereby immune control and killing on the cancerous cells.

## 3. CTLA-4 inhibitor

### Ipilimumab 

Ipilimumab (Yervoy) is a human cytotoxic T-lymphocyte antigen 4 (CTLA-4)-blocking antibody (Table 1). It originally received FDA priority review in 18 August 2010, based on the results from the pivotal MDX010-020 trial for unresectable, late-stage melanoma [28], approved by the FDA on 28 March 2011. On 24 July 2017, the FDA further expanded its approval for unresectable or metastatic melanoma in adults and pediatric patients (12 years and older) based on an open-label, single-arm trial; the approved dose was 3 mg/kg, intravenously, every 3 weeks, for a total of four doses (Category 2A).

On 1 October 2015, ipilimumab in combination with nivolumab received FDA approval for BRAF V600 wild-type unresectable or metastatic melanoma based on CheckMate-069, a randomized, phase 2 study [29] (Category 1). Thereafter, it received expanded approval in combination with nivolumab for unresectable or metastatic melanoma regardless of BRAF mutation status in 2016, per Phase 3 CheckMate-067 trial data [30] (Category 1). Ipilimumab was further expanded by the FDA as an adjuvant treatment of cutaneous melanoma in patients with pathologic involvement of regional lymph nodes (stage IIIA, > 1 mm nodal involvement, IIIB, and IIIC (with no in-transient metastasis) who have undergone complete resection and total lymphadenectomy based on the randomized CA 184-029 study on 18 October 2015 [31] (Category 2A).

In renal cell carcinoma (RCC) with clear cell histology, the FDA approved the use of ipilimumab with nivolumab for previously untreated advanced RCC, with intermediate- or poor-risk RCC, regardless of PD-L1 status based on CheckMate-214, an open-label, randomized (1:1) study [32,33] (Category 1). Patients were risk stratified for RCC by international metastatic database consortium (IMDC) prognostic score and region. In the same CheckMate-214 study, this combination was studied in a favorable-risk group with similar overall survival compared to sunitinib; this combination can be used in relapse and stage IV RCC patients as a subsequent therapy after patients have undergone TKI, VEGF or mTOR therapy [32] (Category 2A).

Ipilimumab in combination with nivolumab was granted accelerated approval based on the CheckMate-142 trial with overall response rate (ORR) and duration of response (DOR) treatment of adult and pediatric patients > 12 years old with mismatch repair deficient (dMMR) metastatic colorectal cancer that has progressed following treatment with agents such as fluoropyrimidine, oxaliplatin, and irinotecan [34] (Category 2A).

## 4. PD-1 Inhibitors

### 4.1. Nivolumab 

Nivolumab (Opdivo), the first human IgG4 monoclonal antibody against PD-1, was initially approved by the FDA on 22 December 2014, based on the outcome of CheckMate-037 (Table 2). This trial showed that ORR improved with fewer toxic effects with nivolumab against standard-of-care chemotherapy among patients with advanced, unresectable/metastatic melanoma who progressed following ipilimumab treatment, or a BRAF inhibitor if BRAF mutation positive [35] (Category 1). On 1 October 2015, the FDA expanded its use in combination with ipilimumab (Yervoy in BRAFV600 wild-type unresectable melanoma or metastatic melanoma under certain circumstances such as high LDH and rapid progression of disease based on the CheckMate-067 andCheckMate-069 trials, which showed improved ORR with the combination immunotherapy [30,35] (Category 1). The FDA again expanded the indication of this combination regimen to metastatic melanoma across BRAF mutation status on 23 January 2016 based on the improved progression-free survival (PFS) rate noted in the CheckMate-067 trial [30] (Category 1). Thereafter, based on the improved recurrence-free survival (RFS) rate from CheckMate-238 trial on 20 December 2017, the FDA further expanded its indication in lymph node-positive or metastatic melanoma patients who had undergone complete resection [36] (Category 1). This was the first indication of nivolumab in the adjuvant setting after direct comparison with ipilimumab and also one of the current first-line systemic therapy in patients with recurrent or metastatic melanoma regardless of BRAF V600 mutation status (Category 1).

Nivolumab later received FDA approval on 4 March 2015 for patients with squamous non-small-cell lung cancer (NSCLC) who progressed after platinum-doublet chemotherapy based on the result of the CheckMate-017 trial, with better ORR as well as survival benefit regardless of the PDL-1 expression level [37] (Category 1). On 10 October 2015, the FDA further expanded its use in metastatic non-squamous NSCLC patients who progressed on first-line platinum-based chemotherapy in a similar setting; this study included patients with actionable mutations such as EGFR and ALK mutation who progressed after appropriate target therapy, per the CheckMate-057 trial, which resulted in increased survival and decreased immunotherapy-related toxicity [38] (Category 1).

On 17 August, 2018, the FDA has also approved nivolumab for small-cell lung cancer (SCLC) patients who progressed on platinum therapy and at least one other line of therapy based on the CheckMate-032 trial, a phase 1/2 multi-center, multi-cohort study, which showed an increase in overall response rate and duration of response [39] (Category 2A).

Nivolumab was also studied among advanced renal cell cancer (RCC) with prior anti-cancer therapy (mTOR) in CheckMate-025 with improved overall survival and fewer side effects [33]. This led to its approval on 23 November 2015 (Category 1). In another trial, CheckMate-214, a phase 3 study, a combination of nivolumab and ipilimumab was compared against sunitinib (standard of therapy) for untreated intermediate- or poor-risk advanced RCC patients based on a better response rate and overall survival (OS) with the combination regimen; the FDA approved the combination regimen for this group of patients on 16 April 2018 as first-line therapy [40] (Category 1). This combination immunotherapy is also approved as first-line therapy for the favorable-risk group (Category 2A), and the second-line therapy for relapse or stage IV disease (Category 2A).

The FDA granted accelerated approval to nivolumab on 17 May 2016 for Hodgkin’s lymphoma that has progressed or relapsed post-autologous stem cell transplantation (ASCT), post-transplantation brentuximab vedotin therapy. It is also approved after three or more lines of therapy that include ASCT (Category 2A based on the higher ORR noted with nivolumab in the CheckMate-205 and CheckMate-039 trials) [41]. 

On 10 November 2016, nivolumab obtained another FDA indication for relapsed/refractory metastatic squamous cell cancer of head and neck (SCCHN) that has progressed on standard-of-care platinum-based therapy. This was based on the OS benefit from a phase-3, CheckMate-141 trial [42] (non-nasopharyngeal—Category 1; nasopharyngeal—Category 2B).

Nivolumab was also studied among patients with locally advanced, unresectable or metastatic urothelial cancer who had progressed despite a platinum-based regimen in the CheckMate-275 trial. In this trial, improved ORR and OS benefit of 7 months irrespective of PD-L1 expression was observed. This formed the basis of its accelerated approval on 2 February 2017 for surgically unresectable or metastatic urothelial cancer [43] (Category 2A).

Nivolumab was studied for metastatic colorectal cancer (mCRC) with microsatellite instability-high (MSI-H) or mismatch repair deficient (dMMR) that had progressed on a combination of fluoropyrimidine, oxaliplatin, and irinotecan. The ORR and duration of response to nivolumab in the CheckMate-142 trial lead to its accelerated approval on 1 August 2017 for this group of patients [34] (Category 2A). In the same cohort of patients, a combination of nivolumab and ipilimumab showed better ORR than with nivolumab alone, which led to accelerated approval of the combination regimen for this group of patients on 11 July 2018, provided patients could tolerate this combination [44] (Category 2A).

Nivolumab was also studied for hepatocellular carcinoma (HCC) that was previously treated with sorafenib in the CheckMate-040 trial. The study was an open-label, phase 1/2, non-comparative dose escalation trial because sorafenib was the only approved drug for HCC at the time. Nivolumab was granted approval for this group of patients via an accelerated process on 22 September 2017, based on the ORR from this trial [45] (Category 2A).

### 4.2. Pembrolizumab 

Pembrolizumab (Keytruda) is another human IgG4k monoclonal antibody against PD-1. It received its first approval on 4 September 2014, via an accelerated process based on the objective response rate of 24%, from the clinical trial NCT01295827, in metastatic melanoma patients who are refractory to CTLA-4 therapy and BRAF inhibitor if they have BRAF mutation [46] (Category 2A), (Table 3). The FDA further expanded its approval for previously untreated advanced melanoma regardless of BRAF mutation status on 18 December 2015 (Category 2A), based on the Keynote-006 trial, a phase 3 randomized trial comparing pembrolizumab against ipilimumab (then standard therapy), which resulted in a prolonged OS and PFS with less toxicity than ipilimumab [47]. Based on Keynote-002, the FDA further expanded its use in ipilimumab refractory advanced Melanoma at the same time as it was shown to be superior to the investigator’s choice chemotherapy [48] (Category 2A). On 19 February 2019, the FDA extended the use of pembrolizumab in the adjuvant treatment of lymph node(s)-positive melanoma following complete resection (Category 1) based on the phase 3, EORTC1325/Keynote-054 trial study demonstrating prolonged recurrence-free survival (RFS) [49]. In the case of metastatic melanoma with limited resectability, if there is no disease after resection, it is still indicated as adjuvant therapy (Category 2A).

Pembrolizumab was approved by the FDA on 2 October 2015 for metastatic NSCLC patients who progressed after platinum-based therapy or EGFR- or ALK-targeted therapy and are positive for PDL-1, via accelerated approval based on the randomized, open-labeled, phase II/III study, Keynote-010 trial, showing improved ORR, PFS, and OS compared to Docetaxel in tumors with at least 1% expression of PDL-1 [50]. There were fewer grade 3–4 adverse events in this study, and they noted signals for higher response rates and survival with higher PDL-1 expression > 50%. Earlier in the same year, nivolumab was also approved for advanced squamous cell lung cancer refractory to first-line therapy regardless of PDL-1 expression, as mentioned earlier in this paper. On 24 October 2016, the FDA expanded its approval as the first-line treatment for metastatic non-small-cell lung cancer with high PDL-1 expression (≥ 50%) but no EGFR or ALK mutation (Category 1 and preferred; category 2B if PDL-1 1–49%). This was based on Keynote-024, which was a randomized, open-label, phase 3 trial comparing pembrolizumab against platinum-based therapy for patients with untreated NSCLC with at least 50% PDL-1 expression but no EGFR or ALK mutation. Interim analysis of this trial showed significantly longer OS and PFS, with fewer treatment-related adverse events than with a platinum-based regimen. This trial was stopped early to allow patients who were still on chemotherapy the opportunity to receive pembrolizumab [51]. On 10 May 2017, the FDA further expanded its approval as a first-line treatment in combination with pemetrexed and carboplatin for metastatic non-squamous NSCLC without EGFR or ALK mutation, irrespective of PDL-1 expression (Category 1 and preferred if PD-L1 expression 1–49%; Category 1 if PD-L1 expression is ≥ 50%). This approval was given via an accelerated process based on improved response rates and PFS from 8.9 months (pemetrexed + carboplatin) to 13 months with the triplet regimen, shown by the keynote-021 trial [52]. This combination was granted full approval on 20 August 2018, based on consistent findings from the Keynote-189 trial [53]. On 30 October 2018, the FDA expanded approval of pembrolizumab as a first-line therapy to metastatic squamous NSCLC in combination with carboplatin with paclitaxel/nab-paclitaxel irrespective of PD-L1 status based on the improved ORR, PFS and OS in a patient with pembrolizumab than without from Keynote-407 [54,55] (Category 1 and preferred if PD-L1 expression 1–49%; Category 1 if PD-L1 expression ≥ 50%). On 11 April 2019, the FDA further expanded its approval of pembrolizumab in NSCLC as a first-line monotherapy for patients with stage 3 NSCLC who cannot undergo surgical resection as well as chemoradiation or metastatic NSCLC with PDL-1 expression ≥1% and no EGFR or ALK mutation. This was based on a phase 3 study, Keynote-042, which showed increased OS in comparison to chemotherapy [56] (Category 1).

On 5 August 2016, pembrolizumab was also approved for recurrent or metastatic head and neck squamous cell cancer (HNSCC) patients who progressed on standard platinum-based therapy (non-nasopharyngeal—Category 1; nasopharyngeal and PD-L1 positive—Category 2B as per NCCN guideline). This approval via an accelerated process on based on the increased ORR (16%) and durability of response (DOR) regardless of human papilloma virus (HPV) status, demonstrated in Keynote-012 trial [57]. On 11 June 2019, the FDA extended its indication as a first-line therapy for patients with metastatic or unresectable, recurrent HNSCC, either as monotherapy in patients whose tumor expresses PD-L1 (combined positive score ≥ 1%) or in combination with platinum and fluorouracil (only for nonnasopharyngeal–Category 2A). This was based on the increased OS noted with the pembrolizumab-based regimen either alone or in combination against those with the cetuximab-based regimen in Keynote-048 [58].

Pembrolizumab was approved for refractory adult and pediatric classical Hodgkin’s lymphoma on 14 March 2017. This approval was given via an accelerated process based on the Keynote-087 trial, which was a single-arm phase 2 study (Category 2A). This showed an overall response rate of 69%—of which, 22% had complete remission and the median duration of response was 11 months [59].

The FDA accelerated the approval of pembrolizumab on 18 May 2017 for unresectable or metastatic urothelial cancer patients who progressed on or after platinum-based therapy including those in the adjuvant setting. This was based on improved 3 month OS, with a lower rate of toxicity noted in the Keynote-045 trial [60] (Category 2A). On the same day, it was also approved as first-line therapy for unresectable or metastatic urothelial cancer patients who are ineligible for cisplatin-containing chemotherapy based on the increased ORR noted in Keynote-052 trial [61] (Category 2A). Patients with PD-L1 expression (combined positive score ≥ 10%) had a better response in this trial, and those with < 10% expression had inferior survival and so the FDA released a statement on 29 June 2018, limiting the use of this drug in patients with locally advanced or metastatic urothelial carcinoma who are not eligible for cisplatin-containing therapy and tumors expressing PD-L1 > 10%, or in patients who are not eligible for any platinum-containing chemotherapy regardless of PD-L1 status (Category 2A as per).

On 23 May 2017, pembrolizumab received accelerated approval for adult and pediatric patients with unresectable or metastatic solid tumors with biomarker selected for MSI-H or dMMR who have progressed after the first-line therapy and also do not have satisfactory alternative therapy, irrespective of the location of the primary tumor with tumor-agnotic approval. This approval was based on multiple clinical trials enrolling patients with MSI-H and dMMR solid tumor (Keynote-012, Keynote-016, Keynote-028, Keynote-158, and Keynote-164) showing an overall response rate of nearly 40% [62,63] (Category 2A).

Pembrolizumab was approved as a third-line therapy for recurrent advanced or metastatic gastric/gastroesophageal junction (GEJ) adenocarcinoma patients with PD-L1 expression (combined positive score ≥ 1%) who have progressed on or after two or more prior lines of therapy including fluoropyrimidine and a platinum-based regimen and, if appropriate, HER2/neu-targeted therapy. This approval was given via an accelerated process on 22 September 2017, based on the RR(13%) and DOR from the Keynote-059 trial [64] (Category 2A). From the NCCN standpoint, pembrolizumab can be used in esophageal (squamous and adenocarcinoma) and EGJ adenocarcinoma (Category 2A for second-line or subsequent therapy for MSI-H or dMMR tumors [65,66]; Category 2B for second-line with PD-L1 expression ≥ 10% [67]; Category 2B for third-line or subsequent therapy) [64].

Pembrolizumab was approved by the FDA on 12 June 2018 for recurrent or metastatic cervical cancer progressing after chemotherapy and PDL-1 positive patients. This approval was given via an accelerated process based on the evidence from cohort E of the Keynote-158 trial showing overall response rate of 14% [68] (Category 2A).

On 13 June 2018, the FDA approved pembrolizumab for adult or pediatric patients with refractory or relapsed primary mediastinal large B-cell lymphoma (PMBCL); this approval was given via an accelerated process on based on the improved ORR (45%) noted in Keynote-170 [69] (Category 2A).

The FDA approved pembrolizumab on 9 November 2018 for HCC patients who had previously been treated with sorafenib, based on RR (17%) and its durability from Keynote-224 [70] (Category 2B, Child–Pugh Class A only).

The FDA granted accelerated approval to pembrolizumab on 19 December 2018, for adult and pediatric patients with recurrent or locally advanced or metastatic Merkel cell carcinoma (MCC) as first-line therapy, based on the ORR (56%) from the CITN-09/Keynote-017 trial [71] (Category 2A).

The FDA has approved pembrolizumab in combination with axitinib as first-line treatment for patients with metastatic renal cell cancer (RCC), based on a phase 3 study, Keynote-426, which showed improved OS, PFS as well as ORR [72] (poor and intermediate risk—Category 1; favorable risk—Category 2A).

### 4.3. Cemiplimab 

Cemiplimab (Libtayo) is a human monoclonal antibody against Programmed Death-1 (PD-1). It was approved by the FDA on 28 September 2018 for the treatment of locally advanced, metastatic cutaneous squamous cell carcinoma patients who are not candidates for curative surgery or radiation, based on the high objective response noted [73] (Category 2A) (Table 4) 

## 5. PD-L1 inhibitors

### 5.1. Avelumab 

Avelumab (Bavencio) is a Programmed Death-Ligand 1 (PD-L1)-blocking antibody and this approval was given via an accelerated process on 18 November 2015 based on the JAVELIN Merkel 200 trial for treatment of histologically confirmed metastatic Merkel cell carcinoma in adults and patients 12 years or older whose disease had progressed on or after chemotherapy [74]. On 23 March 2017, the FDA expanded its approval for metastatic MCC for the treatment of adults and pediatric patients 12 years and older including those who have not received prior chemotherapy (Category 2A). Avelumab is the first FDA-approved treatment for metastatic MCC (Table 5).

In 9 May 2017, the FDA granted accelerated approval to avelumab for patients with locally advanced or metastatic urothelial carcinoma whose disease progressed during or following platinum-containing chemotherapy or within 12 months of neoadjuvant or adjuvant platinum-containing chemotherapy based on the JAVELIN Solid Tumor trial [75] (Category 2A, alternative to preferred agent pembrolizumab, which is category 1). On 14 May 2019, the FDA approved avelumab in combination with axitinib (Inlyta) for the first-line treatment of patients with advanced RCC and it is the first FDA approval for an anti-PD-L1 therapy as part of a combination regimen for patients with advanced RCC. This combination approval was based on results showing significant improved median PFS compared with sunitinib on the JAVELIN Renal trial [76] (Category 2A).

### 5.2. Durvalumab 

In 17 February 2016, the FDA granted Breakthrough Therapy designation for durvalumab (Imfinzi) human monoclonal antibody directed against PD-L1, for the treatment of patients with PD-L1-positive inoperable or metastatic urothelial bladder cancer whose tumor has progressed during or after one standard platinum-based regimen on the basis of early clinical data from a Phase I trial [77] (Table 6). On 1 May 2017, durvalumab received accelerated approval for the treatment of patients with locally advanced or metastatic urothelial carcinoma (mUC) who had disease progression during or following platinum-containing chemotherapy, including those who progressed within 12 months of receiving platinum-containing chemotherapy in the neoadjuvant or adjuvant setting with surgery. The FDA granted accelerated approval to durvalumab based on RR and DOR [77] (Category 2A alternative to preferred agent pembrolizumab, which is a preferred agent and category 1).

On 16 February 2108, the FDA approved durvalumab for the treatment of patients with surgically unresectable stage III NSCLC whose cancer has not progressed after treatment with chemoradiation based on a randomized phase III study, the PACIFIC trial, with signficiant improvement in PFS, OS, increased time to distant metastases or death [78] (Category 1).

### 5.3. Atezolizumab 

On 18 May 2016, atezolizumab (Tecentriq) was approved for the treatment of patients with locally advanced or metastatic urothelial carcinoma (mUC) whose disease progressed during or following platinum-containing chemotherapy, or within 12 months of receiving platinum-containing neoadjuvant or adjuvant chemotherapy in the setting of surgical treatment based on the IMvigor210 trial [79] (Category 2A) (Table 7). Later on, atezolizumab received accelerated approval on 17 April 2017 for for the treatment of patients with mUC who were not candidates for platinum-based chemotherapy regardless of PD-L1 expression based on the Phase II IMvigor210 study [79] (Category 2A). The FDA issued a statement on 29 June 2018 (similar to pembrolizumab) stating that atezolizumab should be used only for patients whose tumors express PD-L1 ≥ 5% (tumor-infiltrating cells) or who are not eligible for any platinum-containing chemotherapy regardless of PD-L1 expression (Category 2A).

On 18 October 2016, the FDA approved atezolizumab for the treatment of people with metastatic non-small-cell lung cancer (NSCLC) with progression on platinum-based chemotherapy, and in progression on targeted therapy in patients with EGFR or ALK abnormalities. This approval is based on two trial results, Phase III OAK [80] and Phase II POPLAR [81] (Category 1).

Atezolizumab, in combination with carboplatin, paclitaxel, and bevacizumab, received FDA approval on 6 December 2018 for the initial treatment of people with metastatic non-squamous NSCLC without any driver mutations such as EGFR or ALK based on results from the Phase III IMpower150 study [82] (Category 1). On 18 March 2019, atezolizumab, in combination with carboplatin and etoposide, received approval for first-line treatment in extensive-stage small-cell lung cancer, based on results from the Phase III IMpower133 study [83] (Category 1).

The FDA approved atezolizumab plus paclitaxel on 8 March 2019 for the treatment of adults with metastatic triple-negative breast cancer in people whose tumors express PD-L1 based on data from the Phase III IMpassion130 study [84] (Category 2A).

## 6. Conclusions

The above-mentioned comprehensive summary of checkpoint inhibitors in malignancies based on the current FDA approval and their recommendation strength per NCCN is subject to change based on future review by the FDA and NCCN. As such, the landscape of immunotherapy in malignancy is rapidly evolving and readers are urged to review the FDA labels for accurate information when using these drugs in current clinical practice.

## Figures and Tables

**Figure 1 cancers-12-00738-f001:**
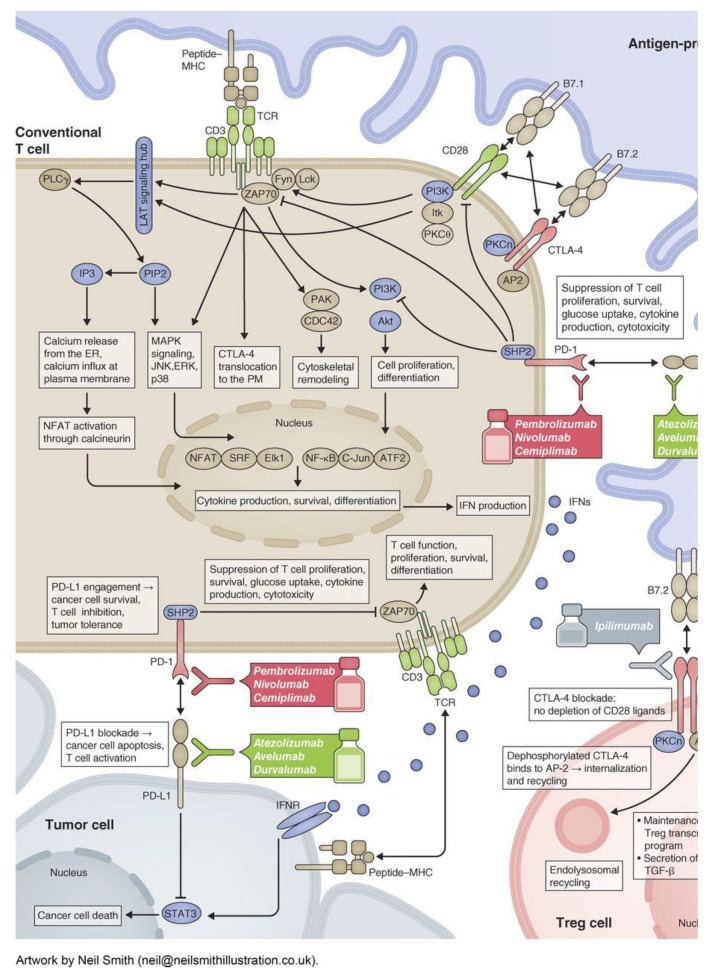
Mechanism of action of all FDA-approved checkpoint inhibitors (published with permission from the *Journal of Cell Biology*).

**Figure 2 cancers-12-00738-f002:**
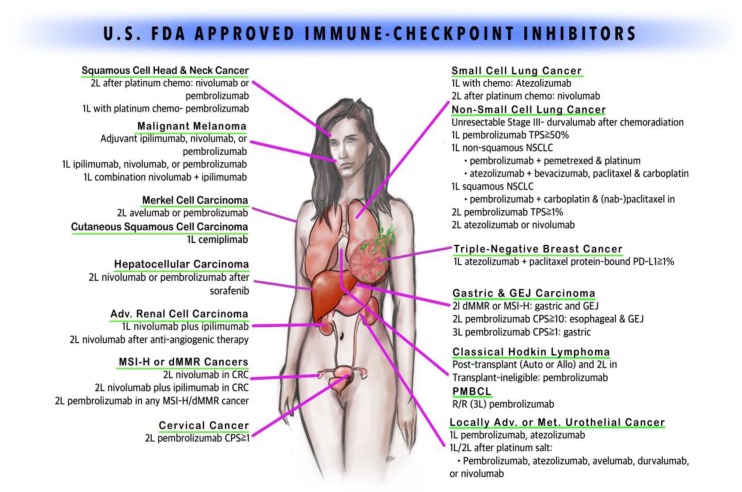
FDA-approved immune checkpoint inhibitors (copyright owned by Raju Vaddepally, et al.).

**Table 1 cancers-12-00738-t001:** Ipilimumab.

Indications	NCCN Guideline Category
Surgically unresectable, stage 3 or 4 malignant melanoma, previously treated or untreated in adults and pediatric patients > 12 years	2A
BRAF V600 wild-type unresectable or metastatic melanoma	1
In combination with nivolumab for unresectable or metastatic melanoma across BRAF status	1
Adjuvant treatment of cutaneous melanoma stage IIIA, IIIB, and IIIC after complete resection along with total lymphadenectomy	2A
In combination with nivolumab, for patients with previously untreated advanced renal cell carcinoma (RCC), relapse and stage IV, with intermediate- or poor-risk RCC, regardless of PD-L1 This combination can be used in relapse and stage IV RCC patients as a subsequent therapy after patients have undergone TKI, VEGF or mTOR therapy	1 2A
In combination with nivolumab for microsatellite instability-high (MSI-H) or mismatch repair deficient (dMMR) metastatic colorectal cancer that has progressed following treatment with fluoropyrimidine, oxaliplatin, and irinotecan in adults and pediatric patients >12 years	2A

**Table 2 cancers-12-00738-t002:** Nivolumab.

Indications	NCCN Guideline Category
Unresectable or metastatic melanoma cancer progressed following treatment with ipilimumab, or a BRAF inhibitor in BRAF mutation-positive patients	1
In combination with ipilimumab for unresectable or metastatic melanoma across BRAF status	1
Lymph node-positive or metastatic melanoma patients who had undergone complete resection	1
Current first-line systemic therapy in patients with recurrent or metastatic melanoma regardless of BRAF V600-mutation status	1
Second line regardless of the histological subtype in non-small-cell lung cancer (NSCLC) in patients who showed progression despite the platinum-based therapy	1
Small-cell lung cancer (SCLC) patients who progressed on platinum-based therapy and at least one other line of therapy	2A
Advanced renal cell cancer (RCC) with prior anti-cancer therapy (mTOR)	1
In combination with ipilimumab, for patients with previously untreated advanced RCC, relapse and stage IV, with intermediate- or poor-risk RCC, regardless of PD-L1 This combination can be used in relapse and stage IV RCC patients as a subsequent therapy after patients have undergone TKI, VEGF or mTOR therapy	1 2A
Hodgkin’s lymphoma that has progressed or relapsed after auto-HSCT and post-transplantation brentuximab vedotin therapy, or three or more lines of systemic therapy that includes auto-HSCT	2A
Recurrent or metastatic squamous cell cancer of head and neck (SCCHN) that has progressed on or after platinum-based therapy (non-nasopharyngeal—Category 1*; nasopharyngeal—Category 2B*)	1* 2B*
Surgically unresectable or metastatic urothelial cancer	A
In combination with ipilimumab for microsatellite instability-high (MSI-H) or mismatch repair deficient (dMMR) metastatic colorectal cancer that has progressed following treatment with fluoropyrimidine, oxaliplatin, and irinotecan in adults and pediatric patients > 12 years	2A
Hepatocellular carcinoma (HCC) previously treated with sorafenib	2A

**Table 3 cancers-12-00738-t003:** Pembrolizumab.

Indications	NCCN Guideline Category
Metastatic melanoma refractory to ipilimumab and BRAF inhibitor with BRAF mutation	2A
Previously untreated advanced melanoma regardless of BRAF mutation status	2A
Adjuvant treatment of lymph node(s)-positive melanoma following complete resection	1
Metastatic melanoma with limited resectability, if there is no disease after resection, as an adjuvant therapy	2A
Metastatic non-small-cell lung cancer (NSCLC) that progressed after platinum-based therapy or, if appropriate, targeted therapy (EGFR/ALK mutation) and positive for PDL-1	1
First-line treatment in patients with metastatic non-small-cell lung cancer with high PDL-1 expression (≥ 50%) but no EGFR or ALK mutation	1 2B if PDL-1 1–49%
First-line treatment in combination with pemetrexed and carboplatin for metastatic non-squamous NSCLC without EGFR or ALK mutation, irrespective of PDL-1 expression	1
First-line treatment in metastatic squamous NSCLC in combination with carboplatin with paclitaxel/nab-paclitaxel regardless of PD-L1 status	1
First-line monotherapy in patients with stage 3 NSCLC who are not candidates for surgical resection as well as chemoradiation or metastatic NSCLC with PDL-1 expression ≥ 1% and no EGFR or ALK mutation	1
For recurrent or metastatic squamous cell cancer of head and neck (HNSCC) patients with progression on standard platinum-based therapy (non-nasopharyngeal—Category 1*; nasopharyngeal and PD-L1 positive—Category 2B*)	1* 2B*
First-line therapy for patients with metastatic or unresectable, recurrent HNSCC either as monotherapy in patients whose tumor expresses PD-L1 (combined positive score ≥ 1%) or in combination with platinum and fluorouracil	2A
Refractory adult and pediatric classical Hodgkin’s lymphoma	2A
Unresectable or metastatic urothelial cancer with progression on or after platinum-based therapy including in the adjuvant setting	2A
First-line therapy for unresectable or metastatic urothelial cancer patients who are ineligible for cisplatin-containing chemotherapy	2A
Locally advanced or metastatic urothelial carcinoma patients who are not eligible for cisplatin-containing therapy and whose tumors express PD-L1 > 10%, or in patients who are not eligible for any platinum-containing chemotherapy regardless of PD-L1 status	2A
Unresectable or metastatic solid tumor patients with biomarker MSI-H or dMMR who have progressed after first-line therapy without satisfactory alternative therapy, irrespective of the location of the primary tumor	2A
Third-line therapy for recurrent locally advanced or metastatic gastric or gastroesophageal junction (GEJ) adenocarcinoma patients with PD-L1 expression (combined positive score ≥ 1%) who have progressed on or after two or more prior lines of therapy including fluoropyrimidine and a platinum-based regimen and, if appropriate, HER2/neu-targeted therapy	2A
Esophageal (squamous and adenocarcinoma) and EGJ adenocarcinoma, subsequent therapy for MSI-H or dMMR tumors; Category 2B for second-line therapy with PD-L1 expression ≥ 10% Category 2B for third-line or subsequent therapy	2A
	2A
Recurrent or metastatic cervical cancer progressing on or after chemotherapy and positive for PDL-1	2A
Refractory or relapsed primary mediastinal large B-cell lymphoma (PMBCL)	2A
HCC patients who had previously been treated with sorafenib	2B
First-line therapy for adult and pediatric patients with recurrent or locally advanced or metastatic Merkel cell carcinoma (MCC)	2A
Combination with axitinib (Inlyta) as first-line treatment for patients with metastatic renal cell cancer (RCC) (poor and intermediate risk—Category 1*; favorable risk—Category 2A*)	1* 2A*

**Table 4 cancers-12-00738-t004:** Cemiplimab.

Indications	NCCN Guideline Category
Metastatic or locally advanced cutaneous squamous cell carcinoma who are not the candidate for curative surgery or radiation	2A

**Table 5 cancers-12-00738-t005:** Avelumab.

Indications	NCCN Guideline Category
Metastatic Merkel cell carcinoma of adults and pediatric patients > 12 years including those who have not received prior chemotherapy	2A
Locally advanced or metastatic urothelial carcinoma patients whose disease progressed during or following platinum-containing chemotherapy or within 12 months of neoadjuvant or adjuvant platinum-containing chemotherapy	2A
Avelumab in combination with axitinib (Inlyta) for the first-line treatment of patients with advanced renal cell carcinoma (RCC) alternative to pembrolizumab (which is the preferred agent)	2A

**Table 6 cancers-12-00738-t006:** Durvalumab

Indications	NCCN Guideline Category
Locally advanced or metastatic urothelial carcinoma patients with disease progression during or following platinum-containing chemotherapy, or whose disease has progressed within 12 months of receiving platinum-containing chemotherapy neoadjuvant or adjuvant, alternative to preferred agent pembrolizumab	2A
Stage III non-small-cell lung cancer (NSCLC) patients for surgically unresectable tumors and whose cancer has not progressed after treatment with chemoradiation	1

**Table 7 cancers-12-00738-t007:** Atezolizumab.

Indications	NCCN Guideline Category
Locally advanced or metastatic urothelial carcinoma with disease progression during or following platinum-containing chemotherapy, or within 12 months of receiving platinum-containing chemotherapy as neoadjuvant or adjuvant therapy	2A
Locally advanced or metastatic urothelial carcinoma patients who are not candidates for platinum-based chemotherapy regardless of PD-L1 expression	2A
Metastatic non-small-cell lung cancer (NSCLC) patients with disease progression during or following platinum-containing chemotherapy who have progressed on an appropriate FDA-approved targeted therapy	1
In combination with bevacizumab, paclitaxel and carboplatin for initial treatment of people with metastatic non-squamous non-small-cell lung cancer (NSCLC) with no EGFR or ALK	1
In combination with carboplatin and etoposide, for the initial treatment of adults with extensive-stage small-cell lung cancer	1
In combination with paclitaxel for adults with unresectable locally advanced or metastatic triple-negative breast cancer in people whose tumors express PD-L1	2A

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
