# Peer review of "Review of Indications of FDA-Approved Immune Checkpoint Inhibitors per NCCN Guidelines with the Level of Evidence"

_cancers, 2020, doi:10.3390/cancers12030738_

Round 1

Reviewer 1 Report

Dear authors,

After reading your article "Review of Indications of FDA Approved Immune Checkpoint Inhibitors per NCCN guidelines with the level of evidence", I find it very educational and well summarized. Nevertheless, I think it should be rechecksed since I have detected several typos, and different ways of writing through the text. Here some examples: 

Line 144 (space missing before reference 30) line 176 space missing

Typo line 151

Line 159 160 restructure

Names of durgs sometimes in capital sometimes not

Line 245 Name os trials in capitals 

Line 348 PD-(48)L1 what does 48 stand for?

Line 380 space missing before Bevacizumab

After rechecking that, I think it is a nice artcile to be considered to be published. 

Author Response

Reviewer 1 (Round 1)

After reading your article "Review of Indications of FDA Approved Immune Checkpoint Inhibitors per NCCN guidelines with the level of evidence", I find it very educational and well summarized. Nevertheless, I think it should be rechecksed since I have detected several typos, and different ways of writing through the text. Here some examples: 

Line 144 (space missing before reference 30) line 176 space missing- spaces corrected

Typo line 151- corrected

Line 159 160 restructure- done

Names of durgs sometimes in capital sometimes not- corrected

Line 245 Name os trials in capitals – corrected.

Line 348 PD-(48)L1 what does 48 stand for? Corrected, abbreviation used instead

Line 380 space missing before Bevacizumab corrected

After rechecking that, I think it is a nice artcile to be considered to be published.

Reviewer 2 Report

This work is a detailed history of the use of immuno-checkpoints inhibitors (ICI). The paper is well written.

The legends of the figures are missing. Several details of each figure are not explained. The figures are not mentioned in the text.

I cannot say whether this work adds something to our knowledge of this topic because I did not know several information reported regarding this story. But I well know the relvance of IC and ICI applications. In my opinion, it seems just a chronical of the use of ICI and it is a really well performed list of events regarding ICI.

From the scientific point of view I do not get any relevant information but I get really important information on when, how and why these ICI have been used and applied to clinic. It appears a paper describing recent events on a really hot topic like chronicle journalists do.

In other words, the ICI are used in clinic, thus this must be based on the clinical evidence (as indicated in NCCN guidlines) that they are working. I understand that the use of one or another ICI (or a combination of different ICIs) can give (and it had given) different results but if this is the case it is more relevant to deeply report, analyze and discuss the molecular mechanisms at the basis of these clinical results rather than listing the events.

Author Response

Reviewer 2 (round 1)

This work is a detailed history of the use of immuno-checkpoints inhibitors (ICI). The paper is well written.

The legends of the figures are missing- labeled appropriately. Several details of each figure are not explained. The figures are not mentioned in the text.

  1. Figure 1- The purpose of the figure is to just give a an idea of the checkpoint inhibitor drugs and interplay with the immune cells and tumor cells. We believe the explaining the complex interplay of immune mechanism and drug interaction is beyond this review article discussion.

I cannot say whether this work adds something to our knowledge of this topic because I did not know several information reported regarding this story. But I well know the relvance of IC and ICI applications. In my opinion, it seems just a chronical of the use of ICI and it is a really well performed list of events regarding ICI. Yes, we agree this paper review is to report the chronological approval of IO drugs and in addition their NCCN strength recommendation which is quite important in clinical practice.

From the scientific point of view I do not get any relevant information but I get really important information on when, how and why these ICI have been used and applied to clinic. It appears a paper describing recent events on a really hot topic like chronicle journalists do.

In other words, the ICI are used in clinic, thus this must be based on the clinical evidence (as indicated in NCCN guidlines) that they are working. I understand that the use of one or another ICI (or a combination of different ICIs) can give (and it had given) different results but if this is the case it is more relevant to deeply report, analyze and discuss the molecular mechanisms at the basis of these clinical results rather than listing the events. Again, as mentioned above reporting the molecular mechanisms was not the intention of this paper.

Reviewer 3 Report

This is a timely review, I would like to ask the authors to check these sentences:

Abstract:

Ipilimumab is the first and only FDA approved CTLA-4 inhibitor based upon its ability to prolong survival in patients with metastatic melanoma.

Correct? there is no doubt that anti-PD1 are more active in melanoma, so this sentence can be misleading for non-experts.

Programmed cell death 1 (PD-1) is an inhibitory transmembrane protein 21
expressed on T cells, B cells, and NK cells

Correct? I know that PD-1 is also expressed in myeloid cels and these cells are crucial for preclinical (and likely clinical anti-PD-1 activity)

chapter 2:

Programmed cell death 1 (PD-1) is an inhibitory transmembrane protein expressed on T cells, B cells, and NK cells.

As above

Author Response

Reviewer 3 (Round 1)

This is a timely review, I would like to ask the authors to check these sentences:

Abstract:

Ipilimumab is the first and only FDA approved CTLA-4 inhibitor based upon its ability to prolong survival in patients with metastatic melanoma.

Correct? there is no doubt that anti-PD1 are more active in melanoma, so this sentence can be misleading for non-experts. Agree with you, I rephrased the sentence to avoid ambiguity. Line 21,22

Programmed cell death 1 (PD-1) is an inhibitory transmembrane protein 21 
expressed on T cells, B cells, and NK cells.

Correct? I know that PD-1 is also expressed in myeloid cels and these cells are crucial for preclinical (and likely clinical anti-PD-1 activity) That’s true, added NK cells.  Line 23,94

chapter 2:

Programmed cell death 1 (PD-1) is an inhibitory transmembrane protein expressed on T cells, B cells, and NK cells.

As above

Extensive changes made to the review paper, considering equitable distribution of spacing, spelling errors, add/delete words, rephrasing sentences for correctness, clarity, delivery, and engagement; and also ensured the citation order is sequence as well.

Line 17- Abbreviations provided for CTLA-4, PD-1

Line 21- removed sentence which to avoid confusion per author’s request.

Line 23, 24, 92- provided more abbreviations, also added MDSC’s per the author’s request which I do agree that PD-1 is expressed in MDSC’s as well.

Line 29, 30- used abbreviations instead.

Line 33- abbreviation provided for ICI

Line 42- rephrased the sentence for clarity and correctness; citations placed at the end of sentence.

Line 57, 58-rephrased sentence

Line 70-73- added a brief synopsis which explains the reminder of the review paper.

Line 71- indicated figure #1, and figure #2 per author’s request as it was missing in the body of paper.

Line 106,107,110-rephrased sentence

Line 115- removed NCCN with every category to avoid repetition from hereafter. And, all the categories are mentioned at the end of sentence for regularity and easy readability.

Line 176,177- deleted the sentence to avoid redundancy.

Line 113,114,173,174- rearranged the sentence for clarity.

Line 229-231- removed to avoid redundancy, stay focused on the

Line 442-4447- added conclusion for reader clarity, proper ending, and closure.

Round 2

Reviewer 2 Report

I understand the authors' and reviewers' point of view. The paper is identical to the first version. I remain of my first opinion. This paper is really good for educational as indicated by the other reviewers.